# Document Registration: Towards Automated Labeling of Pixel-Level Alignment Between Warped-Flat Documents

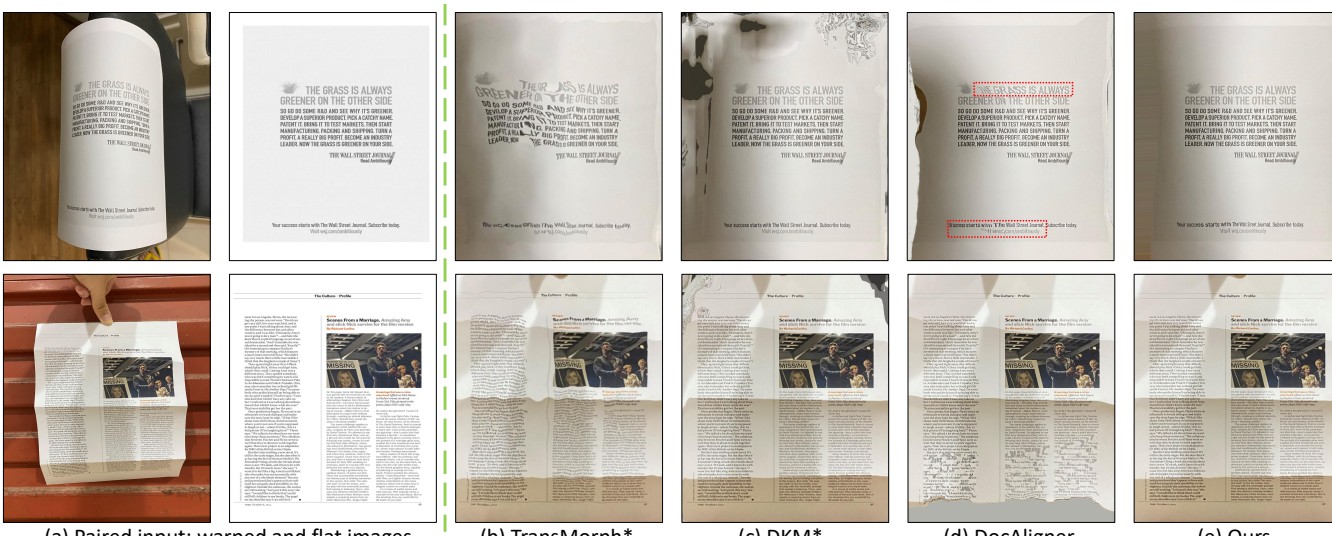

| (a) Paired input: warped and flat images | (b) TransMorph* | (c) DKM* | (d) DocAligner | (e) Ours |

**Figure 1: Comparing results of different image registration methods (TransMorph [5], DKM [10], DocAligner [51]) where we output the dewarping results based on the obtained pixel-level alignments. Since TransMorph and DKM are originally proposed for general images, we re-trained them on document datasets for fair comparison and specified by the symbol "*". Red dotted dashes highlight some poor results in text blocks.**

## ABSTRACT

Photographed documents are prevalent but often suffer from deformations like curves or folds, hindering readability. Consequently, document dewarping has been widely studied, however its performance is still not satisfied due to lack of real training samples with pixel-level annotation. To obtain the pixel-level labels, we leverage a document registration pipeline to automatically align warped-flat documents. Unlike general image registration works, registering documents poses unique challenges due to their severe deformations and fine-grained textures. In this paper, we introduce a coarse-to-fine framework including a coarse registration network (CRN) aiming to eliminate severe deformations then a fine registration network (FRN) focusing on fine-grained features. In addition, we utilize self-supervised learning to initialize our document registration model, where we propose a cross-reconstruction pre-training task on the pair of warped-flat documents. Extensive experiments show that we can achieve satisfied document registration performance, consequently obtaining a high-quality registered document dataset with pixel-level annotation. Without bells and whistles, we re-train two popular document dewarping models on our registered document dataset WarpDoc-R, and obtain superior performance with those using almost 100× scale of synthetic training data, verifying the label quality of our document registration method. The code and pixel-level labels will be released.

## CCS CONCEPTS

• **Applied computing** → *Document management and text processing*; **Document capture**; **Document scanning**; **Annotation**.

## KEYWORDS

Photographed Documents, Document Registration, Image Matching, Document Dewarping, Pixel-Level Alignment

*Conference acronym 'XX, June 03–05, 2018, Woodstock, NY*
© 2018 Copyright held by the owner/author(s). Publication rights licensed to ACM.
ACM ISBN 978-1-4503-XXXX-X/18/06
https://doi.org/XXXXXXX.XXXXXXX

## 1 INTRODUCTION

With the popularity of smartphones, taking photos of documents has become increasingly convenient, which reduces reliance on traditional scanning equipment. But it also brings challenges to the readability of documents due to the severe deformation, resulting in lower OCR performance [29, 38]. To improve readability, document dewarping has been widely studied, aiming to eliminate geometrical deformation as a pre-processing module for OCR [8, 12, 47].

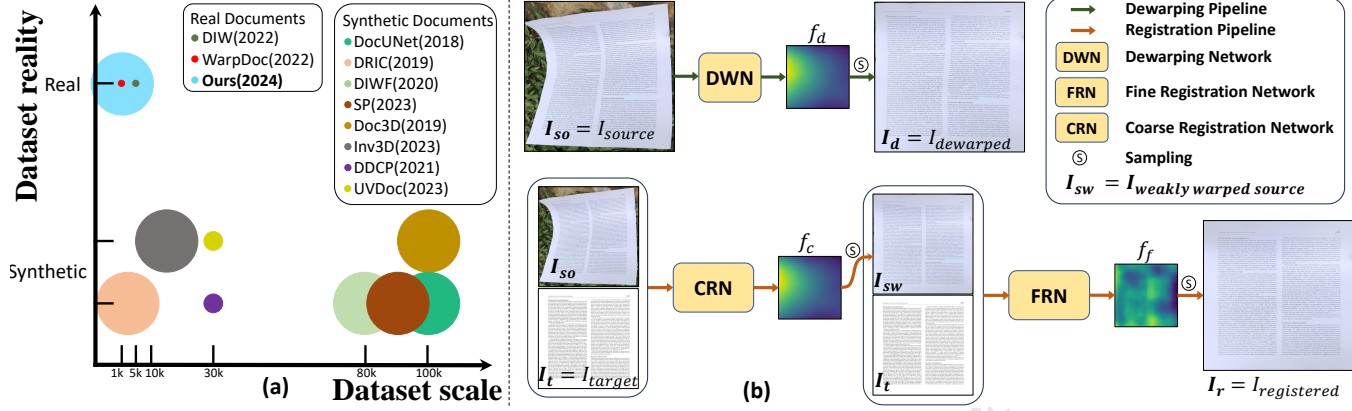

**Figure 2: (a) Summary of document dewarping datasets with three dimensions (sizes, reality and label richness). Small, middle and large circles indicate page-level labels, sparse key point labels, and pixel-level labels, respectively. Our proposed registration method automatically annotates real documents on pixel-level, as shown by the blue circle, see Sec. 2.2 for more details. (b) Comparison of document dewarping (top row, single input) and registration (bottom row, paired input).**

Figure 2 (b) (Top row) shows a typical workflow of current document dewarping models, where the training highly replies on pixel-level annotation between warped-flat documents. Unfortunately, it is costly for humans to manually annotate pixel-level labels on real photographed documents. Thus, most dewarping models are trained on synthetic documents, where the pixel-level labels can be automatically generated during the synthesis process, but the performance of such dewarping models is not satisfied in realistic environment due to the obvious difference of illumination and warping patterns.

We summarize mainstream document dewarping datasets in Figure 2 (a). We can see that most of datasets are synthetic, containing rich labels at pixel level. On the other side, there are only two real documents datasets (i.e., DIW [33] and WarpDoc [49], two small cycles at the left-top corner in Fig. 2(a)), containing very coarse labels only at page level. Such data dilemma motivates us: *can we automatically annotate pixel-level labels for real photographed documents?* In this paper, we propose to automatically annotate pixel-level mapping between real documents and flat counterparts by document registration pipeline, as shown in the bottom row of Fig. 2(b). Through our method, we can generate rich annotation labels on WarpDoc [49] as shown by the large blue cycle[1].

To the best of our knowledge, there is very few research about document registration. One concurrent effort DocAligner [51] presents a dewarping-then-registration framework aiming to align the pixel-level annotations on real datasets, however, the performance is not good as shown in Fig. 1(d). On the other side, dense image registration has been widely studied [32], which usually focuses on natural images, such as building images [35, 53], face images [5, 20], point cloud [4, 19] and street view images [18, 46]. Unlike natural images, photographed documents often contain more complex deformations, diverse layouts, and finer-grained character-level textures. Therefore, directly applying such SOTA dense image registration works into the document images cannot obtain satisfactory results, as shown in Fig. 1(b)(c).

In this paper, we propose a two-stage framework including a coarse registration network (CRN) and a fine registration network (FRN), as shown in the bottom row of Fig. 2(b). The first stage CRN aims at roughly eliminating severe geometric deformation and reducing background interference so that the FRN stage could concentrate on fine-grained distortion registration. Inspired by Inv3D [17], our CRN employs flat documents as templates to iteratively register the corresponding warped documents, obtaining a coarse registered result. Afterward, the FRN applies a multi-scale encoder-decoder to obtain a fine-grained warp flow correction, where we adopt a classification-then-regression [30] idea instead of a common regression block to improve the learning efficiency.

Motivated by the success of pre-training [6, 7, 15], we propose to pre-train the encoder of FRN in this work. In the pre-training task, we impel the warped document to cross-reconstruct the features of its flat counterpart, rather than reconstructing itself in typical Masked Image Modeling (MIM) [15]. This pre-training objective is well suited to document registration tasks as (1) both pre-training and downstream registration rely on a cross-image feature decoder to correlate the pair of input documents for prediction and (2) introducing additional flat documents as a reference helps the pre-trained encoder implicitly understand fine-grained geometric deformations.

To sum up, our contributions are four-fold:

- We propose a document registration pipeline to automatically annotate pixel-level labels on real documents.
- We propose a cross-construction pre-training task for models in document registration.
- Through our proposed document registration method, we enrich the annotations of current real document dataset WarpDoc [49] to form WarpDoc-R.
- Extensive experiments show that we can achieve state-of-the-art registration performance and boost dewarping performance.

---

[1]As no flat documents are released by DIW [33], our pipeline can not be applied.

**(a) Pre-training of Cross-construction**     **(b) Fine-tuning for Document Registration**

Figure 3: Pre-training then fine-tuning pipeline for the FRN stage of document registration. The proposed pre-training task can reconstruct from a masked input and obtain the features of another counterpart. Student Enc. and Student Dec. mean student encoder and decoder respectively, which account for masked image feature extraction and reconstruction. Teacher Enc. denotes teacher encoder to extract features of counterpart. Symbol "//" means gradient stop. The proposed fine-tuning stage contains a shared Enc. (registration encoder) and a Dec. (registration decoder). Finally, the Dec. outputs a warp flow to sample the coarse-dewarped image $I_{sw}$ and obtain the final registered image.

## 2 RELATED WORK

### 2.1 Dense Image Registration: Broad Sense

As a fundamental task in image processing, dense image registration has been widely investigated [3, 31], which is also known as image matching or correspondence [32]. In this paper, we discuss several typical dense image registration methods, including dense geometry matching [10, 35, 36, 42], optical flow estimation [18, 46], and medical image registration [1, 5]. Among them, DKM [10] directly predicts the warp flow between two 3D building scenes at different capture angles and illumination, leading to pixel-level bidirectional mapping relationships, but they are difficult to deal with complex deformation and texture-less margins in document scenes. Distractflow [18] is only suitable for small displacement estimation between adjacent video frames. Transmorph [5] focuses on matching the same landmark feature of specific organs in different domains, which is inapplicable for diverse document layouts. Therefore, it is necessary to design a dedicated model that can handle document registration task.

### 2.2 Document Dewarping: A Data Perspective

In this section, we review document dewarping works from the perspective of datasets, which is summarized in Fig. 2(a). Although photographed documents are convenient to collect, their annotation is extremely expensive to obtain. To this end, early works such DIWF [47], DocUNet [34], DDCP [48], DRIC [25], and SP [23] attempt to synthesize documents by rendering simulated warping shape, illumination, and texture, meanwhile obtaining pixel-level annotations. To improve the reality, some works add real materials into the synthesis process. Doc3D [8] and Inv3D [17] incorporate real warping shapes into the rendering pipeline, while UVDoc [43]

cleverly obtain real lighting rendering by using physically marked ultraviolet ink. However, such synthesize documents still have significant differences from real documents due to the complex and ever-changing real environment, hindering the document dewarping in real application. To overcome this issue, some works [33, 49] directly rely on real documents, but they leverage weakly supervised training since only page-level annotation exists, constraining the dewarping performance. In this paper, we propose document registration pipeline to automatically annotate real documents on pixel-level and obtain a rich labeled real document dataset as shown in the blue circle of Fig 2(a), boosting document dewarping.

### 2.3 Pre-training: Self/Cross-Reconstruction

Pre-training has made great success in computer vision [6, 22, 28], which usually adopts masked image modeling (MIM) [7, 13, 15] for the self-reconstruction. Recently, DocMAE [27] utilizes MIM in document dewarping, demonstrating the effectiveness. However, it is sub-optimal to directly utilize these single-vision pre-trained representations for image registration tasks, because the registration needs to correlate pair input. To address this issue, some works have focused on building dual input pre-training tasks. In dense geometry matching, Pmatch [53] designs a pre-trained encode-decoder framework to reconstruct pair inputs each self. For the optical flow estimation task, Crocov2 [45] establishes pairs of asymmetric branches for the masked original image and the reference image respectively, then reconstructs the original image itself by feature fusion. In this paper, we reformulate the previous MIM to cross-construction in a symmetrical encoder architecture, encouraging the reconstruction of flat image features based on the masked warped source image.

**Figure 4: Architecture of cross-construction pre-training task for FRN, corresponding to the pre-training stage in Fig. 3(a)**

## 3  METHODOLOGY

Our proposed document registration pipeline contains three key components, including CRN, cross-construction pre-training, and FRN in Sec. 3.1, Sec. 3.2 and Sec. 3.3 respectively. As shown in Fig. 2, we first achieve severe deformation elimination and background removal through CRN. Given a warped source document $I_{so}$ and a flat target counterpart $I_t$, CRN can predict the coarse warp flow $f_c$, and then obtain the weakly warped source $I_{sw}$ as the input of following FRN. As for FRN stage, to sufficiently leverage a large number of unlabeled document images and help the training of FRN stage, we first construct a novel pre-training task to pre-train the encoder of FRN, corresponding to Fig. 3(a). Then, we train the entire CRN based on the initialization from pre-training to make it have finer registration capabilities, corresponding to Fig. 3(b). The overall process of FRN can be formalized as: given a weakly warped image $I_{sw}$ and a counterpart $I_t$, FRN can predict the fine warp flow $f_f$, and then obtain the finer registered image $I_r$. Finally, we introduce detailed designs of the loss function in each component.

## 3.1  Coarse Registration Network (CRN)

Unlike the dewarping task, document registration needs to consider dual input. To this end, inspired by Inv3D [17], our first-stage CRN employs flat documents as a template input. Concretely, given a photographed document $I_{so}$ and a counterpart $I_t$, we use two EfficientNet [41] to extract the features of the two images respectively and concatenate them into the GeoTr [12]. In order to adapt to large margin problem [52] in testset, the proposed CRN network is trained in a two-time iteration manner so that the background margin can be removed better. Finally, CRN can predict the coarse warp flow $f_c$, and then obtain the weakly warped intermediate $I_{sw}$. Albeit $I_{sw}$ is perhaps imprecise and still involves weak deformation or introduced artifacts, it can be further corrected by following FRN stage. In this way, we greatly alleviate the learning difficulties caused by severe deformation. The loss function in CRN is defined as the L1-norm between the predicted warp flow $f_c$ and the ground

true warp flow $\hat{f}_c$, formulated as:

$$\mathcal{L}_{crn} = \left\| f_c - \hat{f}_c \right\|_1 \tag{1}$$

Since the architecture of our CRN is largely borrowed from Inv3D [17], we will detail this in Section A of supplementary materials.

## 3.2  Cross-construction Pre-training task

We depict the detailed architecture of cross-construction task in Fig.4. It takes two images $I_{sw}$ and $I_t$ as inputs. cross-construction aims to reconstruct the feature of $I_t$ based on that of $I_{sw}$. A student and a teacher branch is operated in the task. On the one hand, the student branch is made up of a student encoder that extracts the feature of visible token of $I_{sw}$, and a student decoder that predicts the feature of $I_t$. On the other hand, the teacher branch only has a symmetrical momentum encoder updated by exponential moving average (EMA), except for without any masked mechanism. After the pre-training, only the student encoder is used for downstream registration fine-tuning. We will introduce the specific design in four parts.

**Detailed decoupled encoder module**: Since the downstream registration requires progressive registration with multi-scale features, we must pre-train an encoder with multi-scale feature extraction capability. Inspired by the scale-space theory from ROMA [11], we choose a decoupled feature extractor in both student and teacher encoders. Specifically, we employ a parallel ViT [9] and VGG [39] to extract global and local scale features respectively. We formalize this process as:

$$\{\phi_l^t, \phi_l^{sm}\}_{l=1}^5 = \mathcal{E}_\theta(I_t, I_{sw}). \tag{2}$$

where $l = 5$ denotes the global small-scale feature from ViT; $l < 5$ denotes the local large-scale features from each layer of VGG. Note that previous work [51] simply used ResNet [16] to extract multi-level features, we argue that VGG is more suitable for document images with more texture details, as VGG has been proven to handle more precisely localization and have obvious texture bias [14, 37].

**Figure 5: Detailed architecture of decoder in FRN.** $C_g, C_l$ mean correlation volume operation in global and local scope. Operation $up()$ stands for upsampling. Symbol "//" means gradient stop.

**Multi-scale mask mechanism**: Because of the multi-scale properties of the encoder, if only masking ViT without the VGG layers, it will lead to information leakage [13]. Thus, we also mask the VGG layers. In particular, our cross-construction task also adopts a similar block-wise masking strategy in MIM [15]. It initially generates a mask for ViT, then gradually upsamples the mask to higher resolutions for each convolutional layer in VGG. Before each convolution operation, we perform the Hadamard product operation on masks and features.

**Feature fusion module**: In order for student decoders to process these multi-scale features, we fuse these features through the StrideConv operator to upsample and dimensionally transform the different scale feature maps into the smallest scale uniformly. Then all of the features can be easily fused by the addition operator.

**Student decoder module**: We define the results after the two-branch feature fusion module as $\phi_5^{sf}$ and $\phi_5^t$. Since the goal of registration is to obtain the result from $I_{sw}$ to $I_t$. We design a student decoder to recover the features of $I_t$. Specifically, we first incorporate both the mask tokens and visible tokens and feed all tokens into the 4-layer ViT decoder to achieve the prediction of $I_5^t$ features, denoted as $\phi_5^{t'}$.

**Pre-Training task object**: Considering the construction difficulty for dual inputs, we simplify the goal of predicting pixels into predicting features. Therefore, we apply the same mean squared error (MSE) loss from MAE [15], but the study object in our task is to cross-reconstruct flat document's feature $\phi_5^t$ rather than reconstructing pixel. We formulated the MSE loss as:

$$\mathcal{L}_{cc} = \left\| \phi_5^{t'} - \phi_5^t \right\|_2 \tag{3}$$

## 3.3 Fine Registration Network (FRN)

With a pre-trained multi-scale encoder, followed by a decoder in FRN also maintains a multi-scale architecture to correlate two sets of matched features and predict a series of warp flow, as shown in Fig. 5. Inspired by the idea of classification-then-regression [30], our decoder in FRN is divided into two parts: $D_{lc}$ and $D_{op}$. First, at the smallest feature map, we classify the initial mapping position of the $I_{sw}$ through location classifier $D_{lc}$ to obtain the initial warp

flow. Then, a predicted offset is added by multi-scale offset predictor $D_{op}$ to correct the initial warp flow progressively. Given the paired features, we formalize the procedure of the decoder as:

$$f_{l=5}^1 = \mathcal{D}_\theta \{\phi_l^t, \phi_l^{sw}\}_{l=5}^1 \tag{4}$$

where $l = 5$ implies the decoder will start from a small-scale location classifier and then regress the large-scale warp flow offset from a coarse to fine manner. $l = 1$ indicates the finest and largest scale, and $f_{l=1} = f_f$ is the final and finest warp flow as model output. We'll cover two key modules in FRN in two parts:

**Location classifier** $D_{lc}$: Unlike DocAligner [51] and DKM [10], which simply treat multi-scale registration decoder as a serial warp flow regression task from small to large scale, we advocate that the small-scale decoder is much more important than large-scale decoder. The small-scale decoder provides an initial global coarse initialization for subsequent large-scale decoders, which makes it possible to focus further on fine-grained alignment. To design a better small-scale decoder, motivated by HscNet [24] and ROMA [11], we propose to treat the small-scale decoder as a classifier-based initial locator. Specifically, we first calculate the global correlation volume $C_g$ to calculate the vector-by-vector similarity between the two feature maps. Based on $C_g$, we feed the feature map and $C_g$ into a location classifier. we divide the feature map of $l = 1$ into $64 \times 64$ bins. We adopt a 5-layer ViT to predict which bin should be classified for each pixel on the small-scale feature map $\{\phi_{l=5}^t, \phi_{l=5}^{sw}\}$. In this way, flow prediction on a small scale is transformed into a classification task for these bins. Then, according to the classified bins, we convert the positions of these bins into initial warp flow $f_{l=5}$ as the input of the subsequent large-scale decoders.

**Offset predictor** $D_{op}$: With the initial warp flow provided by the small-scale decoder, we further perform offset warp flow $\Delta f_l$ correction under the large-scale decoder. The process is denoted as:

$$f_l = up(f_{l+1}) + \Delta f_l \tag{5}$$

In order to determine the offset value that needs to be corrected, here we apply the previous warp flow $f_{l+1}$ to sample $\phi_l^{sw}$, as shown in Fig. 5(b). In addition, we also calculate the correlation volume between sampled $\phi_l^{sw}$ and $\phi_l^t$. Since the global correlation requires

a large amount of GPU memory, we only calculate the local correlation $C_l$ in the $7 \times 7$ local neighborhood, as shown in Fig. 5(b). With all these prior ($C_l$, Sampled feature, feature map, warp flow), we concatenate and feed them into a convolution-based offset predictor. This process can be formalized as:

$$\Delta f_l = D_{op}(C_l \oplus S(\phi_l^{sw}, up(f_{l+1})) \oplus \phi_l^t \oplus up(f_{l+1})) \quad (6)$$

In such a way, we can progressively correct warp flow to obtain a final warp flow $f_f$.

The loss function for FRN is divided into a localization classification loss for $D_{lc}$ and a regression loss for multi-scale $D_{op}$, denoted as:

$$\mathcal{L}_{frn} = \mathcal{L}_{lc} + \mathcal{L}_{op} \quad (7)$$

$$= -\sum_{c=1}^{C} \hat{p_{ij}}(c) \log p_{ij}(c) + \sum_{l=1}^{L} \left\| f_f - \hat{f_f} \right\|_1 \quad (8)$$

where $\mathcal{L}_{lc}$ is cross-entropy loss, $\mathcal{L}_{op}$ is L1-norm. $C$ is the location bin class index, and L is the scale index. $p_{ij}(c)$ is the probabilities for the ijth pixel in small-scale feature map $\{\phi_{l=5}^t, \phi_{l=5}^{sw}\}$. $f_f$ is fine-grained warp flow.

## 4 EXPERIMENTS

### 4.1 WarpDoc-R: Pixel-level Annotated WarpDoc

Through our proposed document registration method, we enrich the annotations of current the largest real document dataset Warp-Doc [49] to form WarpDoc-R. To this end, we first remove those documents containing occlusions and extreme deformations to ensure a stable registration training process, then we split the remaining 840 documents into training (800 samples, 95%) and test (40 samples, 5%) sets.

During the training, we synthesize severely warped documents and slightly warped documents based on such 800 samples, which are used for the training of CRN and FRN respectively. For severely warped documents, we apply the same image rendering software provided by Inv3D [17] and Doc3D [8] to render 25,000 documents. For slightly warped documents in FRN fine-tuning, we overlay 500 shadow distributions from SD7K [26] with 12,000 warp shapes from DocAligner [51] to simulate photographed documents. During the cross-reconstruction pre-training stage, we first infer 800 raw photographed images on the well-trained CRN, and then the obtained coarsely dewarped documents $I_{sw}$ serve as input for the pre-training task. We also randomly augment $I_{sw}$ by adding deformation and color jitter in the pre-training stage.

Based on the well-trained registration network, we can automatically obtain pixel-level annotations for all 840 pairs of samples, forming WarpDoc-R, which will be released. We can use the training set of 800 samples to train document dewarping model, and the remaining 40 samples are for evaluation of both document registration and dewarping models.

### 4.2 Implementation Details

We train our registration model in three parts: CRN, Cross-Construction pre-training and FRN. For CRN, we train for 300 epochs using a batch size of 8 based on the AdamW optimizer, in which we used the OnceCycleLR scheduler [40] and control the maximum learning

**Table 1: Quantitative comparisons of document registration performance on the WarpDoc-R test set. "*" means the model is re-trained.**

| Method | Venue | MS-SSIM↑ | LD↓ | AD↓ | ED↓ | CER↓ |
|---|---|---|---|---|---|---|
| TransMorph* [5] | MIA'22 | 0.691 | 6.88 | 0.308 | 1835 | 0.546 |
| DKM* [10] | CVPR'23 | 0.752 | 5.04 | 0.106 | 389 | 0.141 |
| DocAligner [51] | Arxiv'23 | 0.830 | 4.09 | 0.0338 | **283** | 0.0997 |
| Ours | - | **0.835** | **3.19** | **0.0313** | 318 | **0.0842** |

**Table 2: Dewarping quantitative comparisons on fixed dewarping model GeoTr (Without pre-segmentation and augmentations) trained by different dataset scales.**

| Dataset | Data type | Dataset scale | MS-SSIM↑ | LD↓ | AD↓ | ED↓ | CER↓ |
|---|---|---|---|---|---|---|---|
| | Synthetic | 0.8k | 0.476 | 12.5 | 0.580 | 1740 | 0.410 |
| Doc3D [8] | Synthetic | 20k | 0.483 | 9.14 | 0.356 | 784 | 0.207 |
| | Synthetic | 80k | 0.528 | 8.88 | 0.306 | 643 | 0.170 |
| Ours | Real | 0.8k | **0.629** | 6.19 | **0.190** | 632 | **0.168** |
| Ours+Doc3D | Real+Synthetic | 0.8+20k | **0.643** | 5.39 | **0.143** | 491 | **0.132** |

rate to 1E-3. The input image resolution is set to $280 \times 280$. For pre-training, we trained for 100 epochs using a batch size of 12 based on the AdamW optimizer. We use 10 epochs as a warm-up with a peak learning rate of 1E-3. Regarding the masking strategy, we follow BEiT [2] to use blockwise masking and set the patch size as $16 \times 16$. For FRN, we set the batch size as 12 to train 80 epochs, and use AdamW as the optimizer along with the initial learning rate 2E-5 and 4E-4 for encoder(pre-trained) and decoder respectively. The learning rate is reduced by a factor of 0.2 after each 30 and 60 epochs. Both pre-training and FRN training adopt $384 \times 512$ as input image resolution. You can find more details on the training and network setting in Section B of the supplementary materials.

### 4.3 Evaluation Metrics

Since we do not have ground-truth of pixel-level mapping labels in our test set of WarpDoc-R, we rely on the dewarping performance to evaluate our registration model. Specifically, we leverage the registration results of our model to dewarp photographed documents, then compare to the flat counterpart to calculate dewarping metrics. In the evaluation of document dewarping models, we directly inputs photographed documents and output dewarped result, which are compared with flat counterpart to calculate metrics.

We employ three widely-used dewarping metrics [8, 34] including similarity, feature alignment, and OCR error rate. For similarity, we adopt MS-SSIM [44] to measure the general similarity between $I_d/I_r$ and $I_t$. For feature alignment, we measure LD (Local Distortion) [50] and AD (Aligned Distortion) [33] by the dense SIFT flow [31]. For the OCR error rate quantization, we select ED (Edit Distance) and CER (Character Error Rate) [21], and view the OCR result of flat document as ground truth.

### 4.4 Results of Document Registration

The registration performance will directly affect the subsequent dewarping task. To evaluate the validity of our proposed registration framework. We conduct both quantitative and qualitative experiments on the proposed benchmarks of WarpDoc-R. Note that

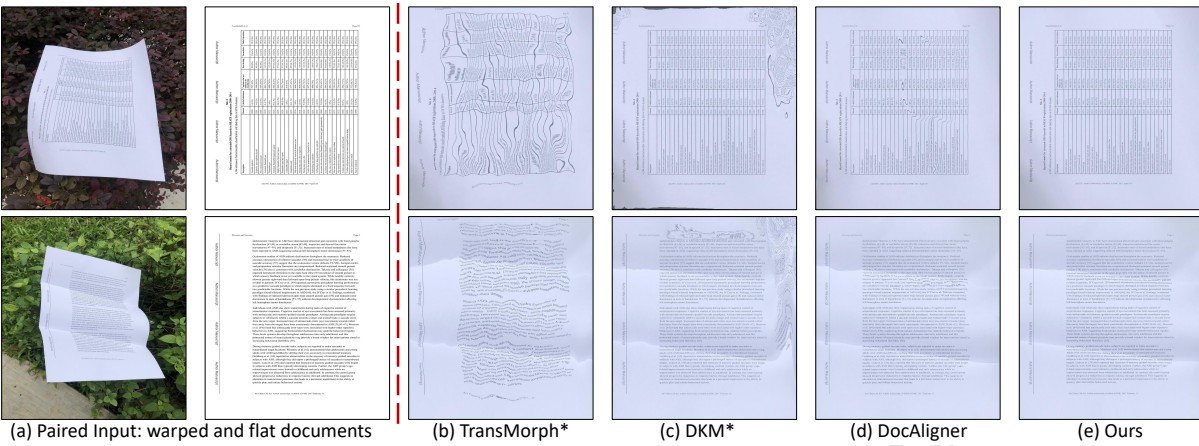

Figure 6: Qualitative comparison of different registration methods (b~e). Our method is superior to existing methods in both character-level alignment and texture-less areas. Zoom-in is recommended for better visualization. Since TransMorph and DKM are originally proposed for general images, we re-trained them on document datasets and specified by the symbol "*".

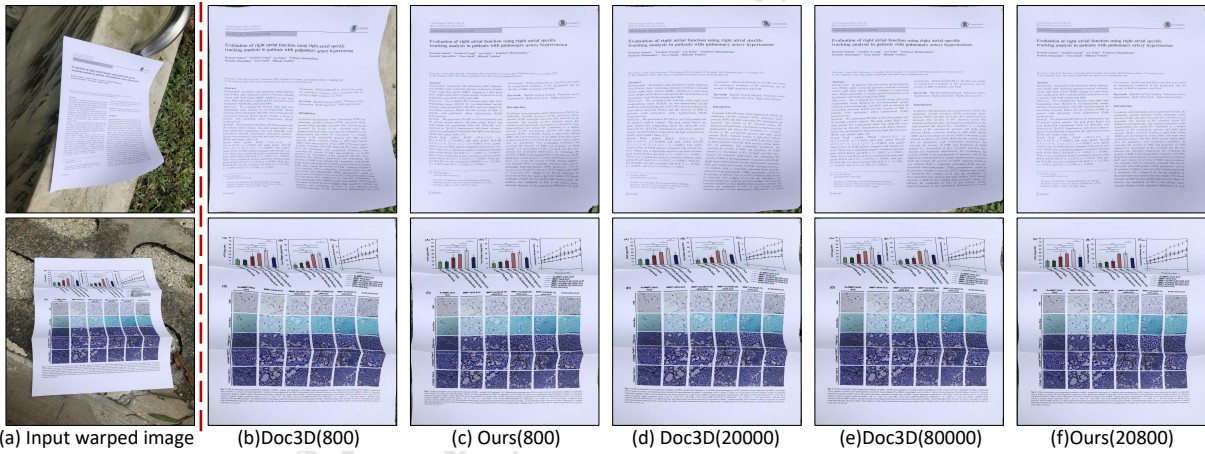

Figure 7: Qualitative comparison of the dewarping model GeoTr [12] trained on different data. Noted that we remove some tricky module like pre-segmentation, random background replacement for fair training comparison.

Transmorph [5] and DKM [10] don't take into account two-stage registration. For a fair comparison, we use $I_{sw}$ from our CRN stage to serve as re-training input for the two works. When evaluating the two works, we only evaluate their fine registration capability. As shown in Tab. 1, our method achieves state-of-the-art performance on most metrics, among which we show significant superiority on LD metrics. Our approach not only outperforms previous state-of-the-art general image registration approaches but also exceeds DocAligner proposed concurrently. As shown in Fig. 6, the proposed method can be well applied to different deformations such as curves or folds, and can be well suited to different texture features in document images, such as table lines, characters, and so on. In addition, our proposed method also be capable of suppressing the error matching for texture-less areas in the margin area, compared with Fig. 6(c). For more visual comparison, you may refer to the Section C of the supplementary materials.

## 4.5 Results of Document Dewarping

Furthermore, we also validate the label quality of our registration method from the perspective of dewarping. Doc3D [8] is currently the most popular and widely used synthetic document dataset, and we attempt to prove through experiments that our small amount of real document data is better than large-scale synthetic documents. Concretely, we re-train two popular document dewarping models (GeoTr [12] and DewarpNet [8]) on our registered WarpDoc-R and Doc3D respectively. As shown in Tab. 2, even with only registered 800 real document samples, we can also obtain superior performance with those using almost 100× scale of synthetic training data (line 3 in Tab. 2). This phenomenon demonstrates the importance of real training samples, that is *fewer high-quality real samples is more valuable than a large amount of low-quality synthetic data.* Due to page space constraints, we have included the Dewarpnet-based dewarping training experiment results in our Section D of

the supplementary materials. We also examine that adding a small number of real documents (0.8k) to the synthetic document dataset (20k) can still improve the dewarping performance again. This also confirms that our registered real documents have extremely high data quality.

## 4.6 Ablation Studies

**Table 3: Ablation study for different components:** $D_{LC}$ denotes the location classifier in FRN, Flow represents the model predict target is warp flow's offset rather than a pixel mapping's offset.

| Network Components | | | | Experimental Results | | | | |
|---|---|---|---|---|---|---|---|---|
| CRN | FRN | $D_{LC}$ | Flow | MS-SSIM↑ | LD↓ | AD↓ | ED↓ | CER↓ |
| ✓ | | | | 0.622 | 6.63 | 0.236 | 721 | 0.374 |
| | ✓ | | | 0.715 | 4.15 | 0.206 | 682 | 0.328 |
| ✓ | ✓ | | ✓ | 0.820 | 3.47 | 0.0466 | 649 | 0.347 |
| ✓ | ✓ | ✓ | | 0.815 | 4.14 | 0.0873 | 647 | 0.317 |
| ✓ | ✓ | ✓ | ✓ | **0.835** | **3.19** | **0.0313** | 642 | **0.293** |

**Table 4: Ablation study on mask ratio and in the cross-construction pre-training stage. We also compare the results of without pre-training or with ImageNet pre-training.**

| Pre-training method | MS-SSIM↑ | LD↓ | AD↓ | ED↓ | CER↓ |
|---|---|---|---|---|---|
| Without | 0.812 | 4.75 | 0.0592 | 724 | 0.302 |
| ImageNet | 0.815 | 3.81 | 0.0490 | 718 | 0.304 |
| 0% mask | 0.833 | 3.38 | 0.0319 | 691 | 0.304 |
| 10% mask | **0.835** | **3.19** | **0.0313** | 642 | **0.293** |
| 25% mask | 0.824 | 3.92 | 0.0415 | 706 | 0.318 |
| 50% mask | 0.821 | 4.50 | 0.0671 | 681 | 0.322 |

In this section, we conduct ablation experiments on key components of the two-stage network and different settings of cross-reconstruction pre-training.

**Network components:** As shown in Tab. 3, we verify the effectiveness of four key components of our proposed registration method, respectively. It can be seen from the first two rows that relying only on single-stage registration cannot achieve satisfactory results, especially on the AD metric. On the basis of the complete two-stage method, in the third and fourth lines, we can either estimate a better warp flow initialization by building $D_{lc}$ classifier, or progressively correct warp flow in multi-scale decoder. The model configuration in the last row enjoys all the above properties, thus achieving state-of-the-art performance.

We also provide an ablation study visualization in Fig. 8. Given a pair of input images in Fig. 8(a), if the CRN stage is removed, relying only on FRN will not be able to fully handle these complex distortions and deformations, and will also cause the registered image to contain some redundant background, as shown in Fig. 8(b). Conversely, if the FRN stage is removed, although the model can mitigate severe deformation to a large extent, it cannot focus well on fine-grained texture alignment, as shown in Fig. 8(c). Only by combining the two-stage registration, as shown in Fig. 8(d), can

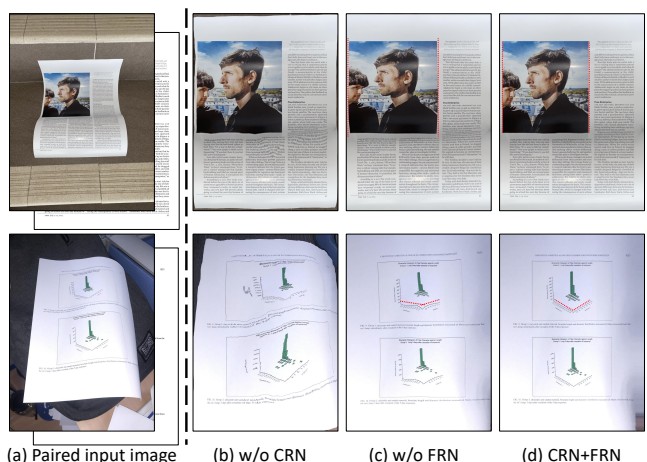

(a) Paired input image    (b) w/o CRN    (c) w/o FRN    (d) CRN+FRN

**Figure 8: Qualitative visual ablation comparison with two-stage registration results of CRN and FRN, respectively.**

we achieve both the capacity of severe distortion mitigation and fine-grained content alignment.

**Pre-training setting:** The masking ratio heavily impacts the difficulty of self-supervised learning. Hence, we adopt diverse mask ratios to explore a suitable value. As shown in Tab. 4, we find that a better registration result can be achieved at a mask ratio of 10%. This is different from the best 70% mask ratio proposed in the original MAE [15]. We consider this is because the document image contains more fine-grained content, and adding an excessively high mask ratio causes a difficult learning process. In addition, as a reference, we also provide results without pre-training and with pre-training on the traditional supervised ImageNet classification task. This shows that the proposed cross-reconstruction strategy has a positive effect on the registration task as a whole.

## 5 CONCLUSION AND FUTURE WORK

In this paper, we propose a document registration pipeline to automatically annotate pixel-level labels on real photographed documents, aiming to solve the data dilemma of document dewarping. Such dilemma lies in the fact that synthetic data has rich labels but lack of reality, while real photographed documents only have page-level annotation. In our document registration pipeline, we construct a coarse-to-fine framework with a coarse registration network (CRN) aiming to eliminate severe deformations then a fine registration network (FRN) focusing on fine-grained features. In the FRN, we leverage a classification-then-regression model to improve the registration performance, furthermore, we propose a cross-reconstruction pre-training task for the encoder of FRN. Through our proposed document registration method, we enrich the labels of one existing real document dewarping dataset to form WarpDoc-R with pixel-level annotation. Extensive experiments demonstrate the high-quality of the pixel-level labels, boosting the document dewarping performance. In the future, we will enlarge the size of real photographed documents to empower the data-centric document intelligence.

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
