# OpenReview forum: "Document Registration: Towards Automated Labeling of Pixel-Level Alignment Between Warped-Flat Documents"
_acmmm.org/ACMMM/2024/Conference — MM2024 Poster_

### Official Review · Reviewer_qKjx · 2024-05-23

**Rating:** 5
**Confidence:** 3

**Summary:**

The paper propose a document registration method. It introduce a coarse-to-fine framework including a coarse registration network (CRN) aiming to eliminate severe deformations then a fine registration network (FRN) focusing on fine-grained feature. The effectiveness the proposed modules are validated. The proposed method can be used to generate high-quality  registered document dataset with pixel-level annotation, which can be used to train dewarping networks for better performance.

**Strengths:**

1、The motivation is clear, and the proposed method can be used to build registered document dataset with higher reality, which is useful in real scenarios.

2、The ablation studies are sufficient to validate the effectiveness the propose CRN and FRN module.

3、The paper is well-written and can be understand easily.

**Limitations:**

1、I notice that the compared DocAligner uses ResNet-50 as its vision backbone, which seem to have smaller parameters and FLOPs than the proposed approach that use a combination of VGG and ViT. It might causes unfair comparison. I would like to know how the proposed approach would perform if VGG and ViT were replaced with ResNet50, the same as DocAligner.

**Suitability:**

2

---

### Official Review · Reviewer_Znba · 2024-05-27

**Rating:** 4
**Confidence:** 3

**Summary:**

This paper introduces a coarse-to-fine framework consisting of a Coarse Registration Network (CRN) to eliminate severe deformations, followed by a Fine Registration Network (FRN) focusing on fine-grained features. The authors employ self-supervised learning to initialize the document registration model, using a cross-reconstruction pre-training task on pairs of warped and flat documents. Extensive experiments demonstrate that the proposed method produces a high-quality registered document dataset with pixel-level annotations. The re-trained two famous document de-warping models on the proposed registered document dataset, WarpDoc-R, achieved superior performance.

**Strengths:**

Methodology:

(I) Propose a document registration pipeline to automatically annotate pixel-level labels on real documents.

(ii) Propose a cross-construction pre-training task for models in document registration.

Experiments:

(I) Using the proposed document registration method, the authors enrich the annotations of current the largest real document dataset WarpDoc to form WarpDoc-R.

(ii) The authors compared the performance of the proposed method with Transmorph, DKM, and DocAligner on document registration tasks. The quantitative and qualitative results show the superiority performance of the proposed method.

(iii) The authors established the effectiveness of the proposed method on dewraping task. The authors used GeoTr network trained on both Doc3D and WarpDoc-R and obtained better results while trained with WarpDoc-R.

(iv) The authors also provide ablation study on the effect of different components of the proposed method.

**Limitations:**

Experiments:

(I) The authors should discuss with few examples where methods fail to registration. Mostly newspapers, magazines since they have complex layouts.

(ii) The authors should discuss the limitation of their methods.

**Suitability:**

3

---

### Official Review · Reviewer_982i · 2024-06-05

**Rating:** 4
**Confidence:** 2

**Summary:**

The paper addresses the challenge of deformations in photographed documents, such as curves and folds, which hinder readability and OCR performance. The authors propose a novel document registration pipeline to automatically align and annotate warped-flat document pairs at the pixel level. This method overcomes the limitations of existing dewarping techniques, mostly trained on synthetic datasets lacking in real-world variations. The proposed pipeline consists of a coarse-to-fine framework, comprising a Coarse Registration Network (CRN) to eliminate severe deformations and a Fine Registration Network (FRN) focusing on aligning finer details of the documents. Additionally, the paper introduces a cross-reconstruction pre-training task for the encoder of FRN. Through this pipeline, the authors enrich the labels of an existing real document dewarping dataset to form WarpDoc-R with pixel-level annotation.

**Strengths:**

Paper Strengths:

- The overall idea of the paper is intriguing. The authors propose an innovative approach to address the data dilemma in document dewarping by introducing a document registration pipeline.
- The proposed method aligns well with the scope of the conference, presenting a significant contribution to the field.
- The combination of CRN and FRN effectively handles both severe and fine-grained deformations, showcasing the versatility of the approach.
- This comprehensive methodology demonstrates thoroughness in addressing different aspects of the problem, making the proposed solution robust and well-rounded.
- The implementation details are provided with sufficient clarity to allow for the reproduction of the results, ensuring transparency and reproducibility of the research.

**Limitations:**

While the paper presents several strengths, in my opinion, it also exhibits some limitations.

- One of its primary contributions is the creation of the enhanced WarpDoc-R dataset, derived from only 840 real document samples. Although the results show promise, the relatively small dataset size might restrict the generalizability of the findings. Expanding the dataset or showcasing results on additional datasets like DIW could bolster the robustness of the study's conclusions.

- The experiments are conducted in controlled settings using the WarpDoc-R dataset. Real-world testing on a diverse set of documents with different types of deformations, occlusions, and lighting conditions could provide a more comprehensive evaluation of the model's robustness and practical applicability

- While the authors justify their choice of VGG over ResNet as a backbone network, stating it's more suitable for document images with intricate texture details, an ablation study varying the backbone architecture could offer deeper insights into the model's performance sensitivity to different architectures.

- The pipeline relies on flat document counterparts, which are not always available, this could be a limitation of this work.

- minor issues such as typos, like 'ground true' instead of 'ground truth' in line 428, should be rectified

**Suitability:**

3

---

### Meta-Review · Area_Chair_vvy3 · 2024-07-03

**Recommendation:** Accept (Poster)
**Confidence:** 4

**Metareview:**

The paper presents a well-rounded, innovative approach to document registration and dewarping, addressing significant limitations in existing methods. The thoroughness in methodology and experimental validation, along with the comprehensive and reproducible details provided, make it a valuable contribution to the field. While there are some limitations regarding dataset size and real-world applicability, these do not overshadow the strengths and potential impact of the work. The reviewers’ concerns have been adequately addressed in the rebuttal, leading to a consensus towards acceptance. Therefore, I recommend acceptance of this paper.